# Basic Mechanical Properties of Duplex Stainless Steel Bars and Experimental Study of Bonding between Duplex Stainless Steel Bars and Concrete

**DOI:** 10.3390/ma14112995

**Published:** 2021-06-01

**Authors:** Qingfu Li, Yunqi Cui, Jinwei Wang

**Affiliations:** School of Water Conservancy Engineering, Zhengzhou University, Zhengzhou 450001, China; lqflch@zzu.edu.cn (Q.L.); wangjinwei11@163.com (J.W.)

**Keywords:** experimental study, stainless steel rebar, mechanical performance, bond behavior

## Abstract

In recent years, as a result of the large-scale use of stainless steel bars in production and life, people’s demand for stainless steel bars has increased. However, existing research information on stainless steel bars is scant, especially the lack of research on the mechanical properties of duplex stainless steel bars and the bonding properties of duplex stainless steel bars to concrete. Therefore, this paper selects 177 duplex stainless steel bars with different diameters for room temperature tensile test, and then uses mathematical methods to provide suggestions for the values of their mechanical properties. The test results show that the duplex stainless steel bar has a relatively high tensile strength of 739 MPa, no significant yield phase, and a relatively low modulus of elasticity of 1.43 × 10^5^ MPa. In addition, 33 specimens were designed to study the bonding properties of duplex stainless steel bars to concrete. In this paper, the effects of concrete strength, duplex stainless steel reinforcement diameter, the ratio of concrete cover to reinforcing steel diameter, and relative anchorage length on the bond stress were investigated, and a regression model was established based on the experimental results. The results show that, with the concrete strength concrete strength from C25 to C40, the compressive strength of concrete increased by 56.1%, the bond stress increased by 27%; the relative anchorage length has been increased from 3 to 6, the relative anchorage length has doubled, and the bond stress has increased by 13%; and, the ratio of concrete cover to reinforcing steel diameter increased to a certain range on the bond stress has no significant effect and duplex stainless steel reinforcement diameter has little effect on the bond stress. The ratio of concrete cover to reinforcing steel diameter from 3.3 to 4.5 and the bond stress increased by 24.7%. A ratio of concrete cover to reinforcing steel diameter greater than 4.5 has no significant effect on the bond stress, with the average bond stress value of 20.1 MPa. The duplex stainless steel bar diameter has little effect on the bond stress for the diameters of 12 mm, 16 mm, 25 mm duplex stainless steel bar, and their average bond stress is 19.9 MPa.

## 1. Introduction

The corrosion of steel is a common durability problem for reinforced concrete structures, especially for offshore, harbor and hydraulic structures, and bridges in coastal areas, which are under harsh environmental conditions, such as long-term wet or alternating wet and dry conditions. The corrosion of steel bars not only affects the normal use of the structure, but it also shortens the service life of the structure and even directly endangers the safety of the structure. Repairing these corroded structures not only requires huge capital investment, but it also generates a lot of construction waste, which, in turn, causes a series of environmental problems. Therefore, countries around the world attach great importance to the study of steel corrosion and its protection technology. Currently, the primary measures for improving the durability of reinforced concrete structures and preventing premature and rapid corrosion of reinforcing steel are [1,2,3]: use high-performance concrete, increase the thickness of the concrete protective layer, seal the concrete surface coating, mix with steel rust inhibitors, use epoxy-coated steel bars, cathodic protection of steel bars, use corrosion-resistant composite steel bars, etc. Practice has proven that the above-mentioned measures have improved the durability of reinforced concrete structures to varying degrees, but they have not fundamentally solved the problem of steel corrosion [4,5]. The stainless steel bars that have emerged in recent years, with their excellent corrosion resistance and mechanical properties, can fundamentally solve the problem of steel corrosion in concrete structures [3,6,7,8,9]. At present, many studies have carried out theoretical and experimental studies on stainless steel bars and stainless steel concrete members [10,11,12]. Research has showed that the strength of stainless steel bars is relatively high, there is no obvious yield stage, and the elastic modulus is relatively low, which can significantly improve the durability of concrete structures, and it has good economic benefits for whole-life projects.

Research on reinforced concrete structures has remained consistent: Yue Liu [13] investigated the bonding properties and bond strength between ultra-lightweight concrete and high-strength concrete; Emanuel Freitas [14] studied the bonding properties between reinforcement and low bond concrete; A. Casanova [15] proposed a new finite element method for simulating the bonding effect of reinforced concrete; Yijie Huan [16] investigated the bonding performance between epoxy coated reinforcement (ECSB) and seawater sea sand recycled concrete; Xueyu Xiong [17] simulated the bonding performance of slowly bonded prestressing tendon (RBT) beam soils and compared the bonding performance of RBT beams with that of deformed steel bars. Kamrul Islam [18] studied the effect of various factors on the bond strength of reinforced concrete; M. Harajli [19] conducted a comparative analysis of bond slip characteristics of reinforcement in plain and fibrous concrete; Dorleta Ertzibengoa [20] studied the bond characteristics of carbon and stainless steel flat bars; Eliene Pires Carvalho [21] investigated the bond strength of thin reinforced concrete; and, Le Huang [22] provided a new mechanical model and simulated the nonlinear bond properties of reinforced concrete using a beam test method.

Most scholars currently use high-performance concrete to study the bonding performance of reinforced concrete. The study of stainless steel reinforcement is also austenitic, as duplex stainless steel reinforcement is not common in daily life; it has only become popular in recent years due to its excellent corrosion resistance and mechanical properties. Therefore, there is a lack of research on the bonding properties of duplex stainless steel reinforcement to concrete [3,23]. In this context, studying the basic mechanical properties of duplex stainless steel bars and the bonding performance between duplex stainless steel bars and concrete can provide a reliable theoretical basis for analyzing the performance of duplex stainless steel bars in concrete structural members, which is necessary for the application of duplex stainless steel bars in the engineering structure. In this context, room temperature tensile tests of duplex stainless steel bars were conducted in this paper in order to determine the basic mechanical properties of duplex stainless steel bars. In this paper, utilizing the literature [24,25,26], the factors influencing the bonding performance of reinforcement to concrete are summarized, so the concrete strength, diameter of duplex stainless steel bars, the ratio of concrete cover to reinforcing steel diameter, and relative anchorage length are used as the main factors influencing the bonding performance of stainless steel bars to concrete. The center pull-out test and the beam test are the most widely used test methods for assessing bonding performance in bonding tests. When compared to the beam test, as an additional compression zone exists that surrounds the rebar in the pull-out test, the pull-out test is less accurate when compared to the beam test, but the center pull-out test is a simple test with relatively low cost, simpler operation, and easier fabrication of specimens [27,28]. Therefore, this paper uses the center pull-out test to study the influence of the four above-mentioned factors on the bonding properties of duplex stainless steel bars to concrete. In this context, the objectives of this study are to determine the basic mechanical properties of duplex stainless steel bars by metal tensile testing in order to investigate the bonding performance of duplex stainless steel bars embedded in concrete using roll-out tests and establish the bond strength equation between duplex stainless steel bars and concrete using a multiple linear regression method. This study has some practical significance, and the results will contribute to the understanding of the basic mechanical properties and bonding performance of duplex stainless steel bars and provide a basis for the development of duplex stainless steel reinforced concrete members in the long term.

## 2. Duplex Stainless Steel Bar Room Temperature Tensile Test

In this paper, the model of 1.4362 duplex stainless steel bars is selected for experimental research. Its China code is 022Cr23Ni4MoCuN. Table 1 shows the chemical composition of duplex stainless steel bars.

### 2.1. Specimen Design

Ensuring that the mechanical properties and bonding properties of duplex stainless-steel bars can meet the requirements of reinforced concrete structures is the prerequisite for the application. The mechanical properties include yield strength, tensile strength, elongation after break, and elastic modulus. According to GB/T 228.1-2010 “Room temperature tensile test method for metallic materials” [29], room temperature tensile tests were performed on duplex stainless steel bars of different diameters to obtain the yield strength, tensile strength, elongation after break, and elastic modulus of duplex stainless steel bars. Using proportional specimens, Table 2 shows the duplex stainless steel bar room temperature tensile test specimen design.

### 2.2. Experimental Method and Procedure

In accordance with GB/T 228.1-2010 “Room temperature tensile test method for metallic materials” in the determination of the original cross-sectional area of duplex stainless steel bars, and then they are marked by continuous dots.

The test was loaded by the strain rate control, and the strain rate was selected as 0.00025/s. A pre-tension force was applied before the test in order to eliminate the influence of the inaccurate fixture alignment and the bending of the specimen in the test. The pre-load rate was 10 MPa/s, and it stops when the pretension reaches 4% of the yield force. After preloading, a pre-calibrated electronic extensometer is installed on the specimen and then the loading is started according to the set loading procedure. After the specimen yields, the extensometer is quickly removed and then the specimen is stretched according to the set loading procedure until the specimen breaks.

### 2.3. Analysis of Experimental Results

#### 2.3.1. Analysis of the Fracture Position of Duplex Stainless Steel Bars

The final breaking position of the tensile specimens is different. The breaking point of some specimens is very close to the upper and lower clamps, and the breaking point of some specimens is in the middle. Through the investigation and statistics of 177 tensile samples, it is found that the sample is more likely to break near the identification symbol if there is a steel bar identification symbol in the parallel length range of the sample, as shown in Figure 1; the fracture position does not have this rule when there is no steel bar identification symbol on the sample or if the steel bar identification symbol is located in the range of the fixture.

When this phenomenon occurs, it is considered that the cross ribs of the duplex stainless steel ribs have played a strengthening role. The normal cross-rib spacing is about 2 mm and the cross-rib spacing at the identification symbol is about 4mm, which is to say, within the influence range of the identification symbol, the relative rib area is one-half of the other parts, forming a weak surface, as shown in Figure 1. When the tensile specimen is under tension, the strengthening effect of the identification symbol on the duplex stainless steel bar is weakened, thereby forming the weak point of the entire tensile specimen, and the fracture naturally occurs near this weak surface.

#### 2.3.2. Stress–Strain Curve Analysis of Duplex Stainless Steel Bars

Figure 2 shows the stress–strain curve of the 16 mm duplex stainless steel bar in the room temperature tensile test. When compared with ordinary hot-rolled steel bars, the stress–strain curve of duplex stainless steel bars has no obvious yield steps and no obvious yield strength. The entire stretching process can be divided into three parts, according to the load-displacement curve of duplex stainless steel bars: elastic phase, strengthening phase, and necking phase: during the initial loading period, the stress–strain curve changes linearly, resulting in elastic deformation, which is called the elastic stage; then, the upward trend of the stress–strain curve slows down, resulting in uniform plastic deformation, which is called the strengthening stage; after the maximum tension, the stress–strain curve drops for the first time, and it drops rapidly, resulting in uneven plastic deformation, which is called the necking stage.

#### 2.3.3. Duplex Stainless Steel Bar Basic Mechanical Properties Index

Perform statistical analysis on the test data, and then obtain the average value of the mechanical indexes of the duplex stainless steel bars of various diameters, as shown in Table 3.

From the data shown in Table 3, it can be seen that the measured tensile strength, elongation after break, elastic modulus, and yield strength of duplex stainless steel bars with diameters of 16 mm to 32 mm are very close to each other. The mechanical properties of the duplex stainless steel bars are stable when compared with 500 MPa grade ordinary carbon steel bars in “Steel for reinforced concrete Part 2 Hot-rolled ribbed steel bars” (GB/T 1499.2-2007) [30]. The measured tensile strength, yield strength, and elongation after break of duplex stainless steel bars are higher, indicating that the duplex stainless steel bars have higher strength and better ductility, as shown in Table 4. When used in concrete structures, it can not only significantly save the amount of steel bars, but also improve the seismic performance of concrete structures.

### 2.4. Recommendations for the Basic Mechanical Properties of Duplex Stainless Steel Bars

In order to facilitate engineering applications and scientific research, the tensile strength, elongation after break, modulus of elasticity, and yield strength of the $\chi^{2}$ method of goodness of fit test of the duplex stainless steel bar are measured because it is a new material with less years of use and a lack of corresponding mechanical indicators. The specific inspection steps are as follows:
Hypothesis that $H_{0}$: the measured data (random variable $f_{i}$) follows a normal distribution.Sort the measured data from smallest to largest and calculate the measured frequency $\eta_{i}$.Calculate the theoretical distribution of the distribution function $F(f_{i})$.
(1)$$F(f_{i})=\frac{1}{\sigma \sqrt{2\pi }}\int_{-\infty }^{f_{i}}\exp\left[-\frac{1}{2}{\left(\frac{f-\mu }{\sigma }\right)}^{2}\right]df$$
(2)$$F(f_{i})=\frac{1}{\xi \sqrt{2\pi }}\int_{-\infty }^{f_{i}}\frac{1}{f}\exp\left[-\frac{1}{2}{\left(\frac{\ln f-\lambda }{\xi }\right)}^{2}\right]df$$
(3)$$\lambda =\ln\frac{\mu }{\sqrt{1+\delta^{2}}}$$
(4)$$\xi^{2}=\ln\left(1+\delta^{2}\right)$$Calculate the theoretical frequency $p_{i}$ and the theoretical frequency $np_{i}$.
(5)$$p_{i}=F\left(f_{i+1}\right)-F\left(f_{i}\right)$$Calculate the $\chi^{2}$ value
(6)$$\chi^{2}=\overset{k}{\underset{i=1}{\sum }}\left(\frac{\eta_{i}-np_{i}}{np_{i}}\right)$$Look up the table, calculate $\chi^{2}(k-r-1)$, and make statistical judgments: if
(7)$$\chi^{2}<\chi_{a}^{2}\left(k-r-1\right)$$
where:*k*: number of groups; and,*r*: estimate the number of parameters, which is 2 in this article.

Subsequently, hypothesis *H*_0_ is accepted, and the statistical analysis is calculated by normal distribution; otherwise, hypothesis *H*_0_ is rejected, and the goodness-of-fit test of the measured test data is performed again.

The measured tensile strength, elongation after break, modulus of elasticity, and yield strength of duplex stainless steel bars of each diameter from 16 mm to 32 mm were tested according to the steps, and the test results regardless of diameter are shown in Table 5. According to the “Unified Standard for Reliability Design of Engineering Structures” (GB 50153-2008) [31], the standard value of elastic modulus is calculated according to the 0.5 quantile value of its probability distribution, and the standard for tensile strength, elongation after break, and yield strength values are calculated according to the 0.05 quantile of their respective probability distributions.

The standard value of elastic modulus is calculated by Formula (8):(8)$$f_{k}=\mu_{f}$$

The standard values of tensile strength, elongation after break, and yield strength are calculated using Formula (9):(9)$$f_{k}=\mu_{f}-1.645\sigma_{f}$$

The standard values of tensile strength, elongation after break, elastic modulus, and yield strength of duplex stainless steel bars calculated by substituting the test data into Formulas (8) and (9), as shown in Table 6.

It can be seen from Table 6 that the standard deviation and coefficient of variation of the tensile strength, elongation after break, elastic modulus, and yield strength of duplex stainless steel bars regardless of diameter are small, and the standard values obtained are 739 MPa and 33.58%, 1.43 × 10^5^ MPa, 513 MPa.

## 3. Bonding Performance Test

### 3.1. Experimental Overview

#### 3.1.1. Specimen Design and Production

The center pull-out test specimen size is relatively small when compared to the beam test, and the production cost is low, the test process is simple to operate, and it is suitable for the production of a large number of specimens to measure the duplex stainless steel reinforcement and concrete bond strength. The concrete strengths chosen were C25, C30, C40, and the concrete mix ratio is shown in Table 7; the duplex stainless steel bar diameters (*d*) were chosen to be 12 mm, 16 mm, and 25 mm; the ratios of concrete cover to reinforcing steel diameter (*c*/*d*) were chosen to be 3.3, 4.5, 5.8, 7.3; duplex stainless steel bar relative anchorage lengths (*l_a_*/*d*) chosen were 3, 4, 5, 6; the test protocol designed the specimens with these four factors as variables; and, the detailed parameters of the specimens are shown in Table 8. Group C30R16T4.5L5 was taken as the benchmark group. A comparative test was carried out with the benchmark group by changing a single variable to keep the rest of the variables unchanged. There are 11 groups of comparative tests.

Figure 3 shows the schematic diagram of the center pullout test. The size of the concrete test block is 160 mm × 160 mm × 160 mm, and the center of the test block is equipped with steel bars, and there are three types of duplex stainless steel bars with diameters of 12 mm, 16 mm, and 25 mm. Taking the impact of local compression on the concrete at the loading end into account, a PVC casing with a diameter of 25 mm and a length of 80 mm is placed in the concrete at the loading end to form a bonding section; therefore, the bonding section between duplex stainless steel bars and concrete is 80 mm long. In addition, the stainless steel bars need to extend out of the concrete since the loading end needs to be placed with pads and loading devices.

#### 3.1.2. Loading Device

The center pull-out test was carried out on a 1000 kN Xinsansi electro-hydraulic servo testing machine, and Figure 4 shows the design test loading situation. During the test, the upper end of the reaction frame is fixed on the fixture of the testing machine, the test block is placed in the hanging basket, and the lower fixture of the testing machine clamps the duplex stainless steel bar that extends out of the test block. The speed of 100 N/s is maintained during the loading process until the end of the experiment.

### 3.2. Analysis of Test Results

#### 3.2.1. Forms of Damage

The duplex stainless steel reinforcement and concrete center pull-out test completed in this paper have two forms of damage: duplex stainless steel reinforcement pull-out damage and concrete splitting damage. The surface shape of the duplex stainless steel bars can significantly affect the form of damage, resulting in a rib height increase; rib spacing can be reduced to increase the bite of concrete and duplex stainless steel bars, in favor of increasing the bond stress. Some studies have shown that [32,33]: in the rib spacing and rib height where the ratio is low, the potential damage near the reinforcing rib splits damage along the key line of concrete between the ribs; in the rib spacing and rib height where the ratio is high, the splitting damage of concrete may occur as well as reinforcement and concrete pull-out damage.

Duplex stainless steel bar pull-out damage

Figure 5 shows the effect of duplex stainless steel bar pull-out damage. The specimen of pull-out damage is loaded to the damage load, the force curve suddenly changes direction and gradually decreases, the duplex stainless steel bar displacement rapidly increases, and the loading end and free end displacement remain synchronized, and the duplex stainless steel bar is pulled out. The surface of the concrete specimen was observed, and no cracks visible to the naked eye were found. When the concrete specimen is cut, it can be seen that the transverse ribs of the bonded section of the stainless steel bars are filled with concrete fragments, while the protrusions on the contact surface of the bonded section of the concrete have disappeared, and obvious shear marks can be seen, indicating that the raised concrete on the duplex stainless steel bar between the ribs was sheared off.

2.Concrete splitting damage

Figure 6 shows the effect of concrete splitting damage. The splitting damage specimen is loaded to the damage load, which is accompanied by a loud sound, the concrete specimen splits into two or three pieces, and then separates from the duplex stainless steel reinforcement. At this time, the force value suddenly decreased, and the test was terminated. Observing the split concrete specimen, it can be found that the convex concrete between the duplex stainless steel ribs in a small area near the loading end is sheared, and the convex concrete between the duplex stainless steel ribs in most of the bonded sections is still in.

#### 3.2.2. Bond Stress

The duplex stainless steel reinforced concrete bond stress is not uniformly distributed over its bond length, and the bond stress is calculated using Equation (10) to facilitate analysis.
(10)$$\tau =\frac{F}{\pi dl_{a}}$$
where:
τ: Bond stress;d: Duplex stainless steel bar diameter (mm);$l_{a}$: The length of the bonded section of duplex stainless steel bars (mm); and,F: Test load.

The bond stress between duplex stainless steel bars and concrete can be obtained while substituting various parameters into Formula (10), as shown in Table 9.

#### 3.2.3. The Influence of Various Factors on the Bonding Performance

The effect of concrete strength on bonding performance

There are three sets of specimens of the duplex stainless steel bar and concrete strength change in the concrete center pull-out test. In the specimens, the diameter of the stainless steel bar is 16 mm, the ratio of concrete cover to reinforcing steel diameter is 4.5, the relative anchorage length is 5, and the concrete strength is C25, C30, C40.

In these three groups of a total of nine specimens, pull-out damage occurred in eight specimens, and the specimens in which splitting damage occurred were caused by improper test operations. It can be assumed that the concrete strength on the specimen damage form basically no effect. During the test, the stainless steel bar cross-rib will produce extrusion pressure on the concrete between the ribs; when the extrusion pressure reaches the limit of the concrete, the concrete in front of the stainless steel bar rib is crushed and the stainless steel bar is pulled out.

The bond stress values for the concrete strengths of C25, C30, and C40 specimens were 17.4 MPa, 20.2 MPa, and 22.1 MPa, respectively, and the compressive strengths of the cubes that were poured simultaneously with the three groups of specimens were 28 MPa, 38.1 MPa, and 43.7 MPa, respectively. Figure 7 shows the curves of the bond stresses of the specimens with the change of the compressive strength of the concrete. It is obvious that, as the compressive strength of concrete increases, the bond stress between stainless steel reinforcement and concrete also increases. From C25 to C30, the compressive strength of concrete increased by 36.1% and the bond stress increased by 16.1%; from C30 to C40, the compressive strength of concrete increased by 14.7% and the bond stress increased by 9.4%. Therefore, if other parameters are kept constant, then the bond stress between duplex stainless steel bars and concrete increases by 14% to 17% for every 10 MPa increase in concrete strength. At the same time, we found that the bond stress was well correlated with the compressive strength of concrete; this is similar to previous studies related to bond strength [25].

2.The effect of duplex stainless steel bar diameter on bonding performance

The duplex stainless steel bars and concrete center pull-out test in the stainless steel bar diameter changes in three groups of specimens: specimens in the concrete strength are C30, the ratio of concrete cover to reinforcing steel diameter are 4.5, and the relative anchorage length are 5; the duplex stainless steel bar diameters were 12 mm, 16 mm, and 25 mm.

There are nine specimens in these three groups, and pull-out damage occurred in six specimens. Among them, splitting damage occurred in three specimens, C30R12T4.5L5-3, C30R25T4.5L5-1, and C30R25T4.5L5-3. This is because, during the test, the stainless steel bar cross-rib will produce extrusion pressure on the concrete between the ribs and, as the diameter of the stainless steel bar increases, the extrusion pressure increases when the concrete between the ribs reaches its limit. Splitting damage occurs when the tensile force on the surrounding concrete exceeds the tensile limit of concrete. Therefore, when the diameter of the duplex stainless steel bars increases, the specimens are more prone to splitting failure.

The values of bond stress for duplex stainless steel bars with diameters of 12 mm, 16 mm, and 25 mm are 19.9 MPa, 20.2 MPa, and 19.6 MPa, respectively. Figure 8 shows the curve of bond stress of the specimen with the change of duplex stainless steel bar diameter. The bond stress basically does not change when the duplex stainless steel bar diameter is from 12 mm to 16 mm, which is due to the small change in duplex stainless steel bar diameter and the influence of experimental errors, which results in slight errors between the experimental results and the theoretical study. However, the bond stress becomes smaller as the diameter of the duplex stainless steel bar becomes larger from 16 mm to 25 mm, which is the same as in previous studies [34]. It can be assumed that, as the diameter of the duplex stainless steel bar increases, the bond stress decreases, but the decrease is not significant.

3.The influence of the ratio of concrete cover to reinforcing steel diameter on bonding performance

The duplex stainless steel reinforcement and concrete center pullout test in the protective layer thickness changes in a total of four groups of specimens: specimens in the concrete strength are C30; stainless steel reinforcement diameter are 16 mm; the relative anchorage length are 5; the ratio of concrete cover to reinforcing steel diameter are 3.3, 4.5, 5.8, and 7.3; and, the four groups of specimens bond stress values are 16.2 MPa, 20.2 MPa, 20.1 MPa, and 20.0 MPa.

These four groups of a total of 12 specimens; out of nine specimens with pull-out damage, splitting damage only occurred in the C30R16T3.3L5 group of three specimens. During the test, the duplex stainless steel bar will produce extrusion pressure on the surrounding concrete, so that the concrete is in a state of tension, as seen in the force schematic diagram presented in Figure 9. From the surface of the duplex stainless steel bar outward to the surface of the concrete, the concrete undergoes gradual decreasing tension. The greater the thickness of the protective layer, the relatively more uniform distribution the tensile force, and the greater the tensile force can be provided. When the ratio of concrete cover to reinforcing steel diameter is 3.3, the maximum tensile force that is provided by the surrounding concrete is small, and the pulling process first reaches the concrete tensile limit, which is, the concrete splitting occurs when the duplex stainless steel bars between the ribs concrete has not been crushed. When the ratio of concrete cover to reinforcing steel diameter reaches 4.5, the maximum tensile force that is provided by the surrounding concrete increases, the pulling process of duplex stainless steel reinforcement inter-rib concrete is crushed when the concrete tensile limit state has not been reached, and pull-out damage occurs.

Figure 10 shows the curve of the bond stress of the specimen with the ratio of concrete cover to reinforcing steel diameter. The image shows that, when the ratio of concrete cover to reinforcing steel diameter is less than 4.5, the bond stress has not yet reached the limit value where the concrete splitting damage occurs; with the increase of the protective layer thickness, the bond stress also increases. At this time, the concrete restraint capacity is low, which leads to cracking of the concrete protection layer so the specimen is prone to splitting damage; when the ratio of concrete cover to reinforcing steel diameter is greater than 4.5, the bond stress does not increase, and it can be considered that, at this stage, increasing the protective layer thickness can no longer improve the bond stress. At this point, the concrete already has a large enough restraint, so pull-out damage occurs in the specimen.

4.Effect of relative anchorage length on bonding performance

There are four groups of specimens in which the anchorage length of the duplex stainless steel reinforcement and concrete center pull-out test changes: the concrete strength is C30; the stainless steel reinforcement diameter is 16mm; the ratio of concrete cover to reinforcing steel diameter is 4.5; the bond stress values of the four groups of specimens are 18.9 MPa, 19.3 MPa, 20.2 MPa, and 21.3 MPa; and, the corresponding relative anchorage lengths are 3, 4, 5, and 6. In these four groups of a total of 12 specimens, pull-out damage occurred in all specimens. This indicates that the concrete around the duplex stainless steel reinforcement can provide a large enough tensile force to make pull-out damage occur in the specimens. Figure 11 shows the curve of bond stress with a relative anchorage length for the specimens. With the relative anchor length from 3 to 4, the bond stress increased by 2.1%; with the relative anchor length from 4 to 5, the bond stress increased by 4.7%; and, with the relative anchor length from 5 to 6, the bond stress increased by 5.4%. Obviously, with the increase of the relative anchoring length, the bonding stress between the duplex stainless steel bar and the concrete increases, and the increase is significantly improved. This experimental result is contrary to some of the literature results [16,25]. There are two main reasons for this: (1) when compared with ordinary steel bars, duplex stainless steel bars have higher strength, no significant yielding phase, and relatively low modulus of elasticity, which greatly enhances the bond strength with concrete; (2) the bonding stress between ordinary steel bars and concrete is unevenly distributed along the anchoring length. When the anchoring length is short, the high-stress area is relatively large, and when the anchoring length is large, the bonding stress in the anchoring area is very uneven and high-stress The area is relatively short. When compared with ordinary steel bars, the bond stress between duplex stainless steel bars and concrete is more evenly distributed along the anchoring length; when the relative anchoring length increases, the mechanical bite force between duplex stainless steel bars and concrete increases, and the bond strength increases. Accordingly, the bond stress increases [34].

## 4. Regression Model of Bond Stress

The bond stress of duplex stainless steel bars is evaluated when considering the following influencing parameters: concrete strength, duplex stainless steel bar diameter, the ratio of concrete cover to reinforcing steel diameter, and relative anchorage length. The bond stress (τ) can be expressed as a function of the above parameters, as shown in Equation (11).
(11)$$\tau =f(\sqrt{f_{c}{}^{\prime }},\;\sqrt{f_{c}{}^{\prime }}d,\;\sqrt{f_{c}{}^{\prime }}c/d,\;\sqrt{f_{c}{}^{\prime }}l_{a}/d)$$

The effects of the four important parameters were combined in order to derive an analytical expression for the bond stress of duplex stainless steel bars using multiple linear regression analysis. Regression analysis is a mathematical and statistical method for dealing with the correlation between variables. It is able to scientifically seek the law of the event and predict its development trend. Multiple linear regression is a method of fitting the relationship between multiple independent and dependent variables. The parameters of the regression model are determined by linearly fitting the relationship between the independent and dependent variables. Although multiple linear regression may not be the most reliable statistical method, it has the characteristics of simplicity of operation, good fitting performance, and the intuitive presentation of results. Equation (12) shows the analytical expression that is derived for the bond stress of duplex stainless steel bars by using multiple linear regression analysis.
(12)$$\tau =\{\begin{array}{cc} -0.9599+\sqrt{f_{c}{}^{\prime }}-0.0078(\sqrt{f_{c}{}^{\prime }})d+0.5358(\sqrt{f_{c}{}^{\prime }})c/d+0.1299(\sqrt{f_{c}{}^{\prime }})l_{a}/d, & c/d\leq 4.5\\ -0.8887+2.9042\sqrt{f_{c}{}^{\prime }}-0.0075(\sqrt{f_{c}{}^{\prime }})d+0.1275(\sqrt{f_{c}{}^{\prime }})l_{a}/d, & c/d>4.5 \end{array}$$
where, τ is the bond stress (MPa); $f_{c}{}^{\prime }$ is the concrete compressive strength (MPa); c/d is the ratio of concrete cover to reinforcing steel diameter; and, $l_{a}/d$ is the relative anchorage length. The form of the regression equation is a segmental function; when the ratio of concrete cover to reinforcing steel diameter is less than or equal to 4.5, the bond stress is mainly affected by four factors: concrete compressive strength, duplex stainless steel bar diameter, the ratio of concrete cover to reinforcing steel diameter, and relative anchorage length. When the ratio of concrete cover to reinforcing steel diameter is greater than 4.5, the ratio of concrete cover to reinforcing steel diameter has no effect on the bond stress, and then the other three factors mainly affect the bond stress.

Analyzing the above regression equation, it is found that the coefficient of $\sqrt{f_{c}{}^{\prime }}$ in the second stage equation is greater than the coefficient of $\sqrt{f_{c}{}^{\prime }}$ in the first stage equation. This is because, in the second-stage equation, the ratio of concrete cover to reinforcing steel diameter is greater than 4.5, at this time, the ratio of concrete cover to reinforcing steel diameter change has no effect on the bond stress change, and it can be considered that the ratio of concrete cover to reinforcing steel diameter acts as $k\sqrt{f_{c}{}^{\prime }}$ (k is a constant) in the second-stage equation, through the superposition of constant coefficients, so that the coefficient in the second-stage equation is greater than the coefficient in the first-stage equation. A good correlation between the experimental and predicted values can be found by comparing the experimental and predicted values, which indicates that the expression used to evaluate the bond stress of the duplex stainless steel reinforcement predicts the experimental results with good accuracy, as shown in Figure 12. Therefore, the proposed bond stress equation for duplex stainless steel bars is considered to be accurate in predicting the bond strength.

## 5. Summary and Conclusions

In this study, 177 duplex stainless steel bars of different diameters were fabricated for room temperature tensile tests, and mathematical methods were used to provide recommendations for the values of the parameters of the duplex stainless steel bars. The bonding performance of duplex stainless steel bars with concrete was tested by constructing 33 specimens, and the effects of four factors on the bonding stress of duplex stainless steel bars, namely concrete strength, duplex stainless steel bar diameter, the ratio of concrete cover to reinforcing steel diameter, and relative anchorage length, were investigated by controlling the variables. Finally, the analytical expression of the bond stress of duplex stainless steel bars was fitted by multiple linear regression. The research presented in this paper can be summarized, as follows.
The tensile process of duplex stainless steel bars is different from that of ordinary steel bars, which can be divided into three stages: elastic stage, strengthening stage, and necking stage. The mechanical properties of duplex stainless steel bars are stable, and their strength is high. Additionally, there is no significant yield strength, but the elastic modulus is low.Duplex stainless steel reinforcement center pull-out specimens have two forms of damage, respectively, pull-out damage and concrete splitting damage. In duplex stainless steel bars in the extraction process, its raised ribs will produce squeezing force on the surrounding concrete matrix. The squeezing force causes tension in the surrounding concrete. When this tensile stress exceeds the tensile strength of the concrete, the specimen cracks internally, which involves cracks developing from the inside out. When the tensile strength of the concrete specimen is small, the internal cracks develop to the surface of the specimen and splitting damage occurs. When the tensile strength of concrete is higher, no cracks are visible on the surface of the specimen, the concrete in front of the reinforcing rib is crushed, and the reinforcing bar is pulled out, at which time the specimen undergoes pull-out damage.The higher the strength of the concrete, the greater the bond stress between the duplex stainless steel reinforcement and the concrete, and the bond stress is proportional to the square root of the concrete strength. The change in the diameter of duplex stainless steel reinforcement has an effect on the damage form of the specimen: the larger the diameter of duplex stainless steel reinforcement the lower the bond stress. The increase in the ratio of concrete cover to rebar diameter can enhance the crack resistance of the specimen: when the ratio of concrete cover to rebar diameter is less than 4.5 and when the concrete has a small tensile strength, the specimen is prone to splitting damage and the bond stress increases with the ratio of concrete cover to rebar diameter; when the ratio of concrete cover to rebar diameter is greater than 4.5 and when the concrete has a higher tensile strength, the specimen is prone to pull-out damage and bond stress remains basically unchanged. Because the duplex stainless steel reinforcement has higher strength, which greatly enhances the bond strength with concrete and duplex stainless steel reinforcement, bond stress is more uniformly distributed along the anchorage length direction, so that, when the relative anchorage length increases, the mechanical bite between the duplex stainless steel reinforcement and concrete increases, and the bond strength increases accordingly.The duplex stainless steel bar bond stress expression that is established by regression analysis can better fit the test value and predicted value for different concrete strengths, the duplex stainless steel bar diameters, the ratio of concrete cover to bar diameter, relative anchorage length, and the formula established, and the test results are in good agreement. It shows that the bond stress formula is scientific and reliable.

In this study, four factors affecting the bond strength of duplex stainless steel bars were considered, and the influence of the surface characteristics of duplex stainless steel bars on the bond strength need to be further considered. We look forward to discussing more influencing factors in future studies. Because only pull-out tests were used in this study to evaluate the bonding behavior of duplex stainless steel bars in concrete, future studies should use other test methods, such as pull-out tests and beam-sample tests, in order to obtain reliable results and develop design models. This paper only considers the bond strength of duplex stainless steel bars with ordinary concrete, and further studies of different types of concrete are needed to establish a more comprehensive relationship between the bonding performance of duplex stainless steel bars in concrete.

## Figures and Tables

**Figure 1 materials-14-02995-f001:**
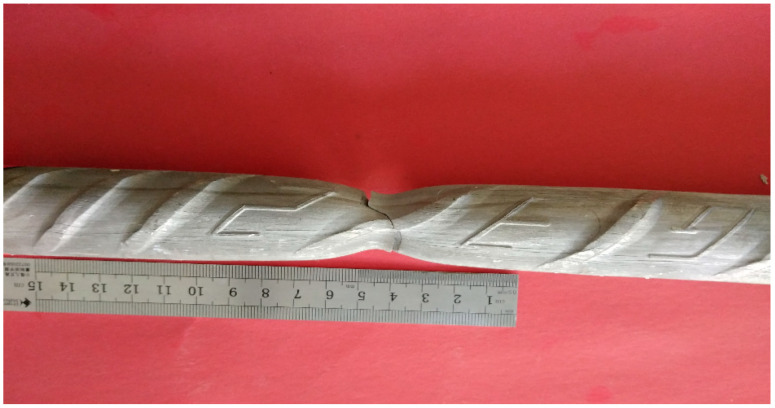
Fracture location of duplex stainless steel bars.

**Figure 2 materials-14-02995-f002:**
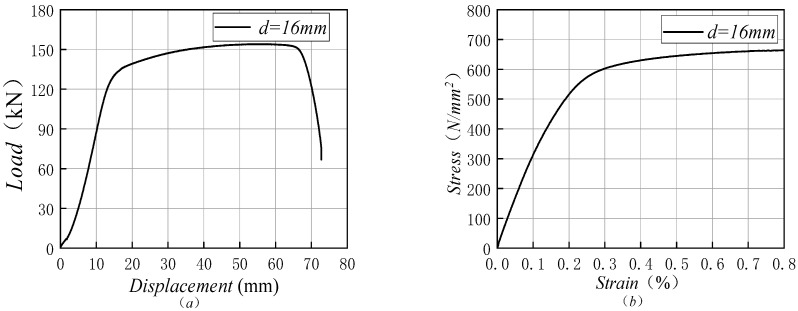
Curve diagram of a duplex stainless steel bar with a diameter of 16 mm. (**a**) is the load-displacement curve; (**b**) is the stress–strain curve.

**Figure 3 materials-14-02995-f003:**
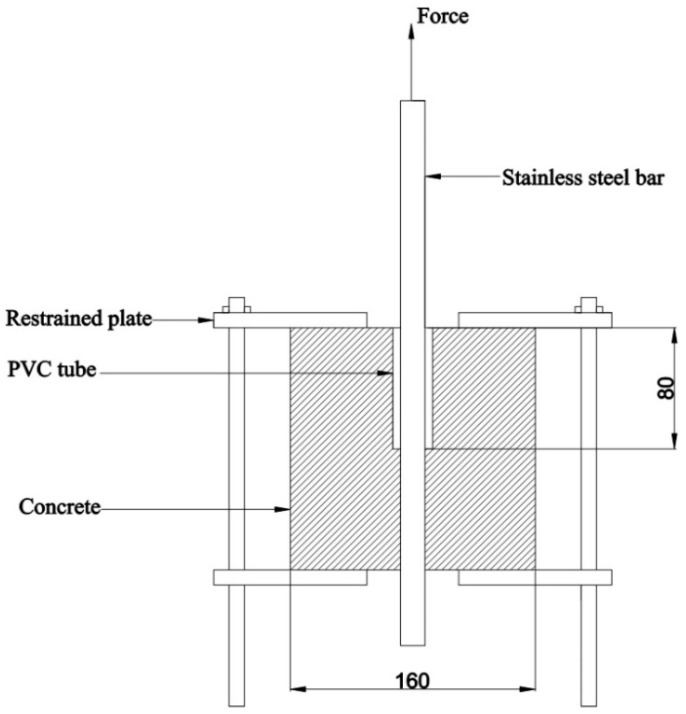
Schematic diagram of the center pull-out test piece (mm).

**Figure 4 materials-14-02995-f004:**
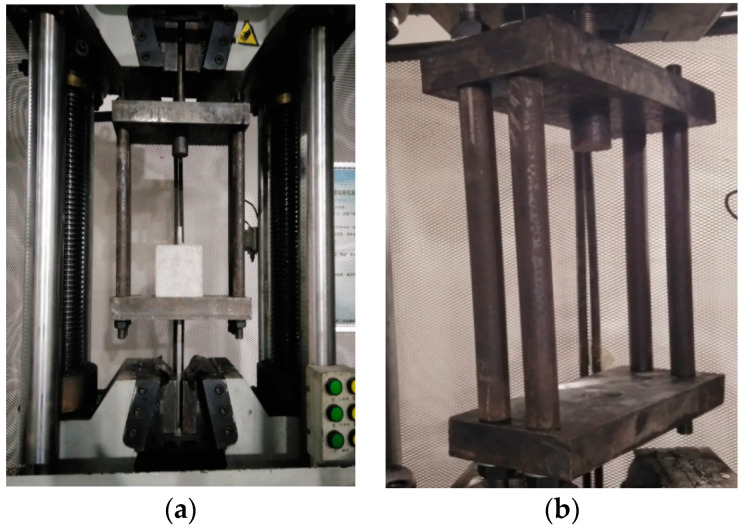
The center pull-out test device includes (**a**) 1000 kN Xinsansi electro-hydraulic servo testing machine, (**b**) reaction force frame structure diagram.

**Figure 5 materials-14-02995-f005:**
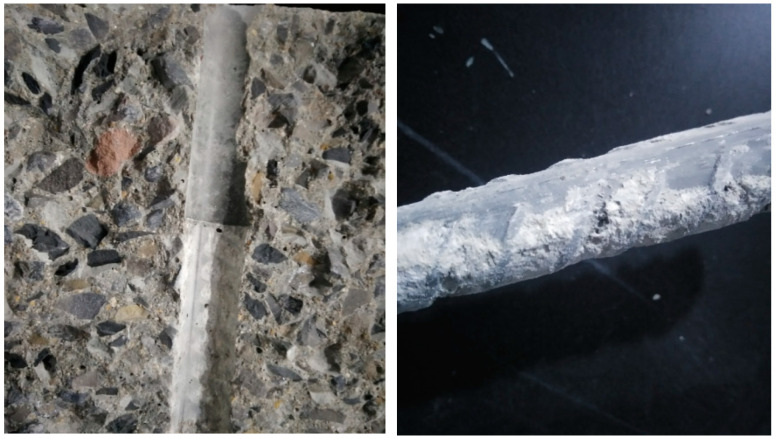
Diagram of pull-out damage.

**Figure 6 materials-14-02995-f006:**
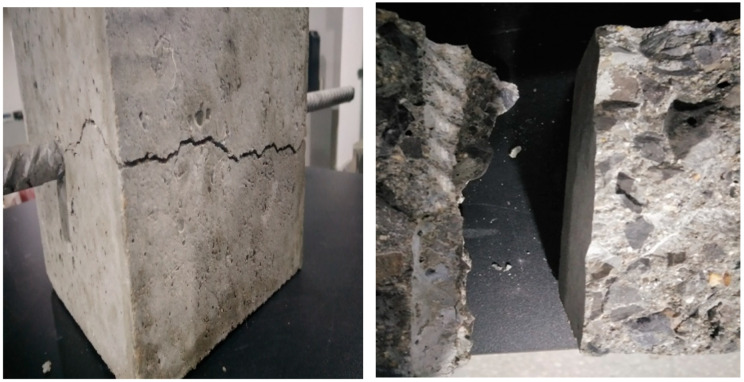
Schematic diagram of cleavage damage.

**Figure 7 materials-14-02995-f007:**
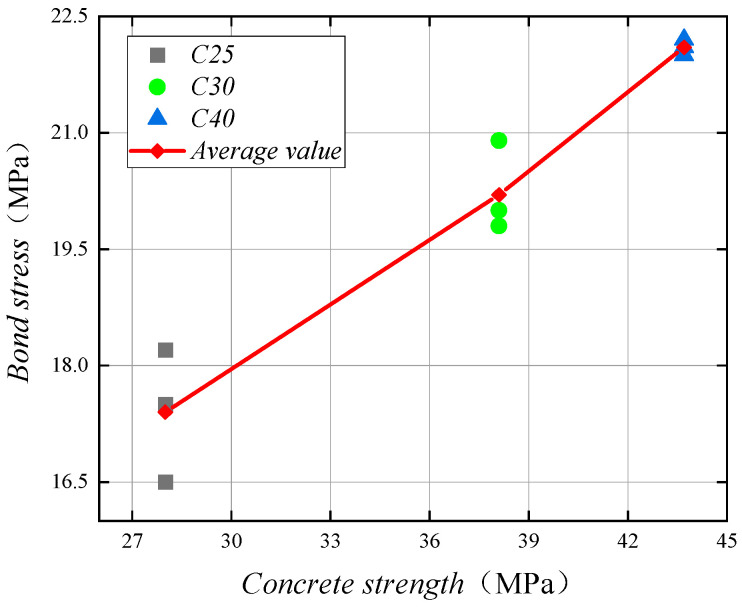
Relationship curve between bond stress and concrete strength.

**Figure 8 materials-14-02995-f008:**
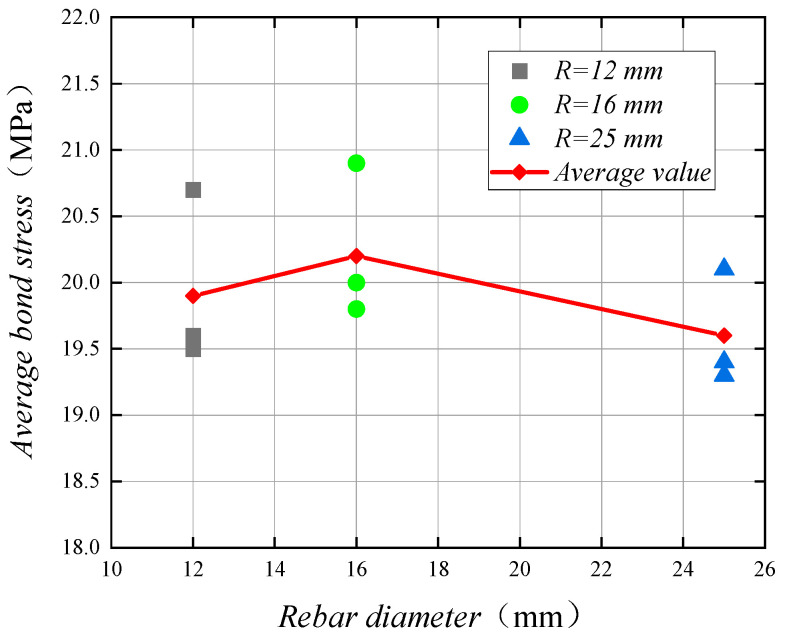
The bond stress and stainless steel bar diameter relationship curve.

**Figure 9 materials-14-02995-f009:**
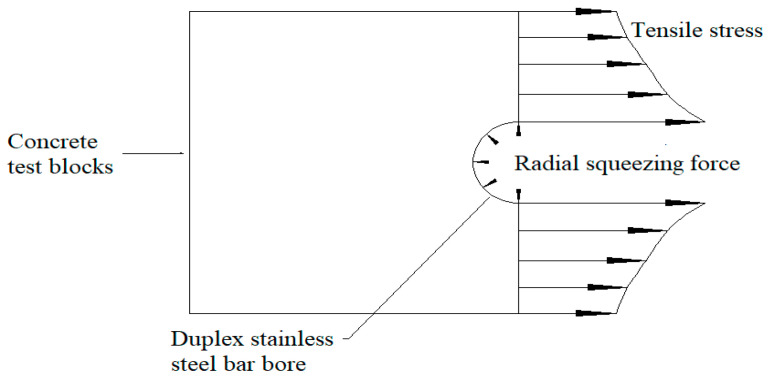
Concrete test block force diagram.

**Figure 10 materials-14-02995-f010:**
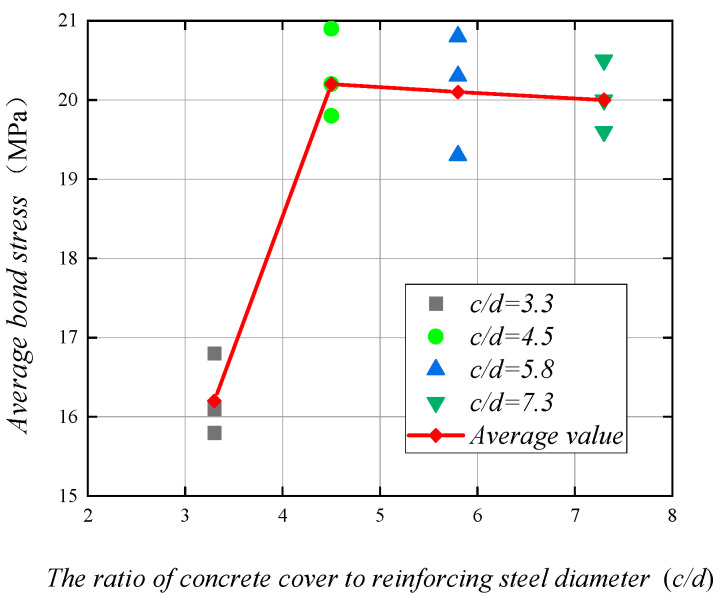
Bond stress and the ratio of concrete cover to reinforcing steel diameter relationship curve.

**Figure 11 materials-14-02995-f011:**
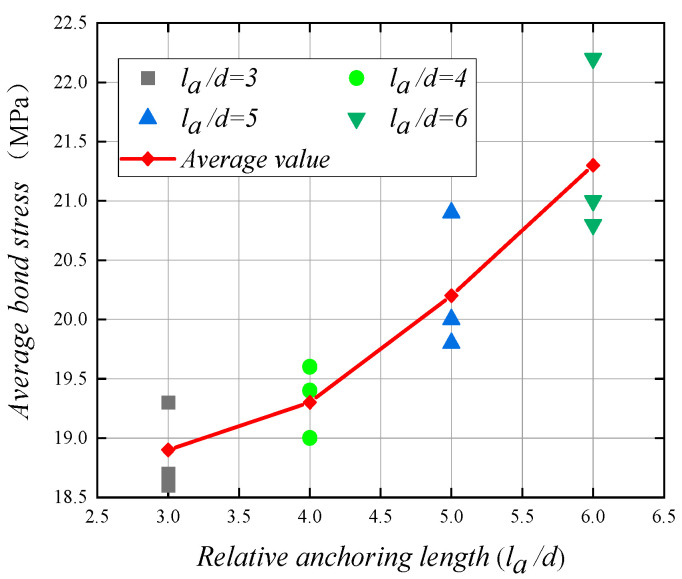
Relationship between bond stress and relative anchorage length.

**Figure 12 materials-14-02995-f012:**
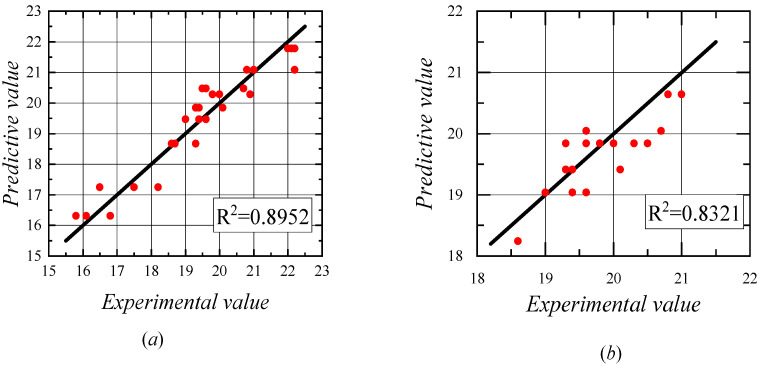
Comparison of the test values and predicted values. (**a**) the ratio of concrete cover to reinforcing steel diameter is greater than 4.5; (**b**) the ratio of concrete cover to reinforcing steel diameter is less than 4.5.

**Table 1 materials-14-02995-t001:** Chemical composition mass fraction of duplex stainless steel bars (%).

Model Number	C	Si	Mn	P	S	Ni	Cr	Mo	Cu	N
1.4362	0.03	1.0	2.5	0.035	0.03	3.0–5.5	21.5–24.5	0.05–0.60	0.05–0.60	0.05–0.20

**Table 2 materials-14-02995-t002:** Specimen design for room temperature tensile test of duplex stainless steel bars.

Number	Diameter/mm	Original Scale Distance/mm	Total Length/mm	Number of Test Roots
L12	12	60	250	32
L16	16	80	300	35
L25	25	125	450	33
L28	28	140	450	39
L32	32	160	500	38

**Table 3 materials-14-02995-t003:** Statistical values of mechanical indexes of duplex stainless steel bars.

Diameter (mm)	Number of Specimens	Tensile Strength (MPa)		Yield Strength (MPa)		Elastic Modulus (10^5^ MPa)		Elongation after Break (%)	
		Average Value	Standard Deviation	Average Value	Standard Deviation	Average Value	Standard Deviation	Average Value	Standard Deviation
12	32	842	9.32	636	23.92	1.56	0.06	33.00	1.43
16	35	768	5.96	531	16.67	1.56	0.08	36.39	2.43
25	33	760	9.21	542	26.94	1.40	0.11	34.13	1.25
28	39	743	9.19	514	17.21	1.38	0.09	39.53	1.21
32	38	748	4.30	527	15.48	1.39	0.13	36.89	0.75

**Table 4 materials-14-02995-t004:** A comparison of the basic mechanical properties of 500 MPa grade plain carbon steel bars and duplex stainless steel bars.

Rebar Type	Tensile Strength (MPa)	Yield Strength (MPa)	Elongation after Break (%)
500 MPa grade ordinary carbon steel bars	630	500	15
Duplex stainless steel bars	750	514	33

**Table 5 materials-14-02995-t005:** Duplex stainless steel bar mechanical indicators normal distribution test results.

Category	Number of Specimens	$\chi^{2}$	$\chi_{\alpha }{}^{2}\left(k-r-1\right)$	Test Results
Tensile strength	177	5.36	5.891	Accepted
Elongation after break	95	12.3	12.69	Accepted
Modulus of elasticity	175	5.62	7.85	Accepted
Yield strength	175	10.33	12.92	Accepted

**Table 6 materials-14-02995-t006:** The mechanical properties of duplex stainless steel tendons performance index standard values.

Indicators	Tensile Strength (MPa)	Yield Strength (MPa)	Modulus of Elasticity (×10^5^ MPa)	Elongation after Break (%)
Average value	755	529	1.43	36.74
Standard deviation	9.82	10.02	0.07	1.92
Coefficient of variation	0.01	0.02	0.05	0.05
Standard value	739	513	1.43	33.58

**Table 7 materials-14-02995-t007:** Concrete mix ratio table.

Concrete Strength	Water-Cement Ratio	Sand Rate (%)	Cement (kg/m^3^)	Water (kg/m^3^)	Sand (kg/m^3^)	Coarse Aggregate (kg/m^3^)	Admixtures
C25	0.68	38	287	195	729	1189	Not used
C30	0.60	37	325	195	696	1184	Not used
C40	0.49	36	398	195	651	1156	Not used

**Table 8 materials-14-02995-t008:** Bonding test specimen grouping table.

Number	Concrete Strength	Concrete Compressive Strength, *f_c_*′ (MPa)	Diameterd/mm	*c*/*d*	*l_a_*/*d*	Specimen Side Length/mm	Number of Test Pieces
C25R16T4.5L5	C25	28	16	4.5	5	160	3
C30R16T4.5L5	C30	38.1	16	4.5	5	160	3
C40R16T4.5L5	C40	43.7	16	4.5	5	160	3
C30R12T4.5L5	C30	38.1	12	4.5	5	120	3
C30R25T4.5L5	C30	38.1	25	4.5	5	250	3
C30R16T3.3L5	C30	38.1	16	3.3	5	120	3
C30R16T5.8L5	C30	38.1	16	5.8	5	200	3
C30R16T7.3L5	C30	38.1	16	7.3	5	250	3
C30R16T4.5L3	C30	38.1	16	4.5	3	160	3
C30R16T4.5L4	C30	38.1	16	4.5	4	160	3
C30R16T4.5L6	C30	38.1	16	4.5	6	160	3

Note: In the number, C30 means the concrete strength is C30, R16 means the duplex stainless steel reinforcement of 16mm diameter, T4.5 means the ratio of concrete cover to reinforcing steel diameter (*c*/*d*) is 4.5, L5 means the relative anchorage length (*l_a_*/*d*) of duplex stainless steel reinforcement is 5.

**Table 9 materials-14-02995-t009:** The test results of bond strength between duplex stainless steel bars and concrete.

Number	Form of Damage	Damage Load (KN)	Bond Stress (MPa)
C25R16T4.5L5-1	Pull-out	73.3	18.2
C25R16T4.5L5-2	Pull-out	66.3	16.5
C25R16T4.5L5-3	Pull-out	70.3	17.5
C30R16T4.5L5-1	Pull-out	84.0	20.9
C30R16T4.5L5-2	Pull-out	79.6	19.8
C30R16T4.5L5-3	Pull-out	80.4	20.0
C40R16T4.5L5-1	Splitting	88.4	22.0
C40R16T4.5L5-2	Pull-out	88.8	22.1
C40R16T4.5L5-3	Pull-out	89.1	22.2
C30R12T4.5L5-1	Pull-out	44.0	19.5
C30R12T4.5L5-2	Pull-out	46.9	20.7
C30R12T4.5L5-3	Splitting	42.0	19.6
C30R25T4.5L5-1	Splitting	207.1	20.1
C30R25T4.5L5-2	Pull-out	179.4	19.3
C30R25T4.5L5-3	Splitting	190.6	19.4
C30R16T3.3L5-1	Splitting	61.9	15.8
C30R16T3.3L5-2	Splitting	69.9	16.8
C30R16T3.3L5-3	Splitting	64.7	16.1
C30R16T5.8L5-1	Pull-out	81.4	20.3
C30R16T5.8L5-2	Pull-out	77.7	19.3
C30R16T5.8L5-3	Pull-out	83.4	20.8
C30R16T7.3L5-1	Pull-out	80.3	20.0
C30R16T7.3L5-2	Pull-out	78.9	19.6
C30R16T7.3L5-3	Pull-out	82.4	20.5
C30R16T4.5L3-1	Pull-out	45.0	18.7
C30R16T4.5L3-2	Pull-out	46.6	19.3
C30R16T4.5L3-3	Pull-out	44.9	18.6
C30R16T4.5L4-1	Pull-out	66.2	19.6
C30R16T4.5L4-2	Pull-out	62.4	19.4
C30R16T4.5L4-3	Pull-out	61.1	19.0
C30R16T4.5L6-1	Pull-out	101.3	21.0
C30R16T4.5L6-2	Pull-out	107.1	22.2
C30R16T4.5L6-3	Pull-out	100.3	20.8

## Data Availability

The data used to support the findings of this study are included within the article.

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
