# Peer review of "Basic Mechanical Properties of Duplex Stainless Steel Bars and Experimental Study of Bonding between Duplex Stainless Steel Bars and Concrete"

_materials, 2021, doi:10.3390/ma14112995_

Round 1
Reviewer 1 Report
This paper investigates the mechanical properties of duplex stainless steel bar, including direct tensile test and bond strength. Although results obtained by the present study are interesting and the tests are well-designed, major revision is necessary to improve the structure of the manuscript, as follows:
- Abstract: Please remove the word “corrosion” at the beginning of the abstract, which is not the main topic of this study.
- Abstract: Please add more quantitative results at the end of the abstract.
- Page 1, Lines 41-43: please add references.
- Page 1, Line 45: explain more about “good bonding properties with concrete”. Is there any previous study for this type of rebar regarding the bond test? If yes please add references and explain the novelty of the present study. If no, please revise the text.
- Page 2, Lines 67-69: Please add references and also revise the sentence. Highlight the previous research gaps and novelty of the present work.
- Page 2, Lines 76-77: Please explain the section”…through the literature..”. Is there any previous study for this type of rebar regarding tests conducted in the present study?
- Page 2, Lines 80-81: It is important to mention here that the pull-out test is less accurate as compared to the beam test, as an additional compression zone exists surrounding the rebar in the pull-out test, which affects the results.
- General comment: last paragraph of the introduction should accurately explain the objectives.
- Page 2, Line 85-86: based on which code (classification standard)?
- General comment: As this paper concentrates on bond properties in a new type of rebar, rebar surface deformation (rib spacing, rib height, angle, etc.) should be mentioned for all rebars with different diameters. This also affects the bond failure mechanism.
- Page 4, Line 134: It will be useful to show the stress-strain curve of (typical curve) an ordinary hot-rolled steel bars in Fig. 2(b) for comparing the results.
- Table 3: why the standard deviation is so high for tensile strength and yield strength. It can be due to the high number of repetitions. Why this number of repetitions?
- Page 5, Line 154: Regarding tensile strength, the standard deviation is not small. Please revise this sentence.
- Page 5, Lines 155-159: It is useful to provide a table and perform a comparative study between the results of the present work with ordinary carbon steel bars from the literature.
- General comment: The reviewer recommends separating the section of “results” from the section of “experimental program”. The current version of the manuscript combined these two sections, which need to be revised.
- Page 7, Section 3.1: Please provide a table and mention concrete compositions for all grades including cement weight, water weight, aggregate weights, and admixtures. This is very important.
- Table 6: please mention concrete compressive strength (fc) in the table for all concrete grades.
- Table 6: please mention rebar deformation details for all diameters.
- Section 3.1.2: Please mention the loading rate (mm/min) for the pull-out test.
- Section 3.2.1: rebar deformation significantly affects the bond damage mechanism. Please use the following references to explain your results:
[-] Mousavi, S. S., Guizani, L., & Ouellet-Plamondon, C. M. (2020). Simplified analytical model for interfacial bond strength of deformed steel rebars embedded in pre-cracked concrete. Journal of Structural Engineering, 146(8), 04020142.
[-] Wu, C., & Chen, G. (2015). Unified model of local bond between deformed steel rebar and concrete: Indentation analogy theory and validation. Journal of Engineering Mechanics, 141(10), 04015038.
- 5: It shows the concrete powder at the front face of the ribs. Please mention details of the rebar deformation to discuss the results. Similar comment for Fig. 6.
- Page 10, Line 282: Parameter “A” is explained in the text, while this parameter does not exist in Eq. (10). Please check.
- Page 11, Lines 310-312: this is not clear enough. Please revise the sentence. Moreover, previous studies severally explained that bond strength is directly proportioned to the square root of the concrete compressive strength.
- 12, Lines 331-334: previous studies confirmed that increasing the rebar diameter reduces the bond strength. Please justify your results as compared to the literature for other types of rebar.
- Table 7: please explain the reason for the splitting failure mode of specimens. For instance, specimens with c/d=3.5 have lower concrete confinement, resulting in concrete cover splitting. Please discuss the failure mode of specimens C40R16T4.5L5-1, C30R12T4.5L5-3, C30R25T4.5L5-1, and C30R25T4.5L5-3 in the text.
- General comment: the reviewer recommends using “concrete cover-to-rebar diameter ratio” instead of “relative protective layer thickness”. Use the words commonly mentioned in this field.
- The results of Fig. 11 need to be more explanations. Previous studies reported that as the db/ld increases, the bond strength increases. Please compare your results with the literature.
- (12): please change this equation by normalizing bond strength with the square root of the concrete compressive strength, which was commonly reported by the literature and concrete codes. Moreover, do not use a negative coefficient for c/db ratio for c/db higher than 4.5. You can remove this parameter or mention that c/db has no impact on the bond for this range.
- Section 4: use previous predicting equations for the bond strength to check with your experimental database. The authors should pay more attention to the literature in this field.
- Please provide a section before the section of “Conclusions” to highlight the limitations of the present study to be considered for future works.
Author Response
Please see the attachment, thank you.

Reviewer 2 Report
The paper presents a series of information on experimental study of basic mechanical properties and bonding force of duplex stainless steel bars
From the analysis of the information presented in the article, I found the following:
- The paper presents a series of results that may be of interest to the scientific community:
- The title should be completed so as to result that stainless steel bars are part of the concrete structure;
- The introductory part should be improved taking into account other bibliographic sources. Also, at the end of the introduction, the structure of the paper and the objective of the research must be presented more clearly;
- The research methodology is not properly explained and in this sense the DOE (design of experiment) should have been used from the beginning of the research;
- The technology for obtaining the test pieces must be explained;
- The processing of experimental data and the obtaining of the two mathematical models must be completed so that their adequacy can be demonstrated;
- Experimental data are inadequately analyzed and more emphasis should be placed on establishing the causes that led to their obtaining;
- The discussion part needs to be substantially improved, as the novelty of the results obtained in the research is not highlighted in this section, compared to other current results in the field;
- The conclusions should be more concrete and include a series of information on the practical possibility of using the research presented, as well as future research directions.
Author Response
Please see the attachment, thank you.

Reviewer 3 Report
Dear authors the paper in general is well written. However in my opinion some correction have to be made.
- In table 1 is MO should be Mo. Good would be to present if authors have made tests on one steel the real chemical composition. Very good would be if the chemical composition would be in comparison to standard. The standard should also be defined.
- Table 3 and 4 correct ( ).
- I propose to look closer to the point 4 (regression model …) In my opinion in this form authors should or remove this part or extend it.
- There is lack of discussion concerning all results as a point in paper. After it should be made short part conclusions – this can be in points.
Author Response
Please see the attachment, thank you.

Round 2
Reviewer 1 Report
The authors appropriately improved the structure of the manuscript.
Reviewer 2 Report
The authors revised their manuscript according to my suggestions. Thus the manuscript can be accepted for publication.